# 🐕 Orthus: Autoregressive Interleaved Image-Text Generation with Modality-Specific Heads

Siqi Kou [1] [*]  Jiachun Jin [1]  Zhihong Liu [1]  Chang Liu [1] [*]
Ye Ma [2]  Jian Jia [2]  Quan Chen [2]  Peng Jiang [2]  Zhijie Deng [1]

## Abstract

We introduce Orthus, a unified multimodal model that excels in generating interleaved images and text from mixed-modality inputs by simultaneously handling **discrete text tokens** and **continuous image features** under the **AR** modeling principle. The continuous treatment of visual signals minimizes the information loss while the fully AR formulation renders the characterization of the correlation between modalities straightforward. Orthus leverages these advantages through its modality-specific heads—one regular language modeling (LM) head predicts discrete text tokens and one diffusion head generates continuous image features. We devise an efficient strategy for building Orthus—by substituting the Vector Quantization (VQ) operation in the existing unified AR model with a soft alternative, introducing a diffusion head, and tuning the added modules to reconstruct images, we can create an Orthus-base model effortlessly (e.g., within 72 A100 GPU hours). Orthus-base can further embrace post-training to craft lengthy interleaved image-text, reflecting the potential for handling intricate real-world tasks. For visual understanding and generation, Orthus achieves a GenEval score of 0.58 and an MME-P score of 1265.8 using 7B parameters, outperforming competing baselines including Show-o and Chameleon. Our code is available at `https://github.com/zhijie-group/Orthus`.

---

[*] Work done during an internship at Kuaishou Technology. [1] Qing Yuan Research Institute, Shanghai Jiao Tong University [2] Kuaishou Technology. Correspondence to: Zhijie Deng <zhijied@sjtu.edu.cn>.

*Proceedings of the 42nd International Conference on Machine Learning*, Vancouver, Canada. PMLR 267, 2025. Copyright 2025 by the author(s).

## 1. Introduction

Multimodal models have shown promise in image-to-text and/or text-to-image generation, with LLaVA (Liu et al., 2024d;c), Emu2 (Sun et al., 2024c), and NExT-GPT (Wu et al., 2013) as popular examples. These abilities are essential for handling complex real-world understanding and generation problems. Yet, existing approaches can suffer from significant modeling redundancy due to the trivial combination of specialized large models (*e.g.*, CLIP-ViT (Radford et al., 2021), Stable Diffusion (Rombach et al., 2022), and LlaMa (Touvron et al., 2023a;b)). Doing so also undermines the benefits brought by cross-modal learning and introduces considerable inefficiency for both training and inference.

There is ongoing interest in jointly modeling visual understanding and generation with a unified, compact model. One strategy is to map both images and texts to discrete tokens for simple autoregressive (AR) modeling (Liu et al., 2024f; Team, 2024; Wang et al., 2024) (left of Figure 1). However, the image tokenizer, often equipped with a vector quantization (VQ) bottleneck, can cause inevitable information loss and easily lead to suboptimal performance on vision tasks concerning high-frequency details (*e.g.*, OCR and human face generation). Alternatively, recent works, including Transfusion (Zhou et al., 2024) and Monoformer (Zhao et al., 2024) (middle of Figure 1), propose to integrate AR modeling on discrete text tokens and diffusion modeling on continuous image features within a single transformer. Nonetheless, the nature of diffusion modeling to process noisy images (Ho et al., 2020) makes the joint modeling of image-to-text, text-to-image, and more complicated interleaved image-text challenging.

This paper proposes Orthus[1] to bridge the gap. Orthus conjoins *lossless continuous image features* and *the unified, cross-modal AR modeling* by decoupling diffusion from the transformer backbone. This circumvents the noise disturbance and renders the characterization of the correlation between modalities straightforward, making it more suitable for interleaved image-text modeling. Specifically, Orthus

---

[1] Orthus is a loyal two-headed guard dog in Greek mythology.

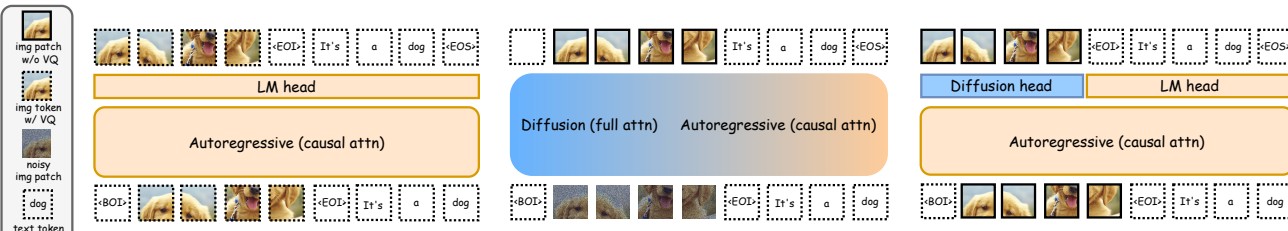

*Figure 1.* Comparison of existing unified multimodal models with Orthus. **Left: Fully AR models** (Liu et al., 2024f; Team, 2024; Wang et al., 2024) convert visual signals to discrete image tokens via vector quantization for joint modeling with text tokens, but this causes information loss. **Middle: AR-diffusion mixed models** (Zhou et al., 2024; Xie et al., 2024) perform next-token prediction for text generation and image patch denoising for image generation, but the involved noise disturbance on images makes the concurrent image-to-text and text-to-image generation challenging. **Right: Orthus** operates in a fully AR manner while circumventing vector quantization and noise disturbance to preserve input information and modeling flexibility.

embeds both discrete text tokens (from an off-the-shelf tokenizer) and continuous patch-wise image features (from a pre-trained variational autoencoder (Kingma & Welling, 2013)) into the same representation space, where an AR transformer is then invoked to model the inter- and intra-modality interdependence. On top of the backbone, Orthus defines two modality-specific heads, with one as the regular *language modeling (LM) head* to predict discrete text tokens and the other as a novel *diffusion head* to craft continuous image features. During inference, Orthus autoregressively predicts the next text token or image patch according to the indication of special transition tokens.

Notably, the investigation into diffusion head for superior image generation draws a striking analogy to the recent masked AR (MAR) approach (Li et al., 2024), yet with a focus shift from image-only generation to mixed-modality one. On the other hand, our Orthus differentiates from MAR and its variant (Yang et al., 2024a) in that it characterizes the correlation with fully AR formulation instead of mask-based modeling, which avoids expensive hyperparameter specification and eases the modeling of interleaved data.

The other important contribution of this work is a super-efficient strategy to build Orthus. Inspired by that Orthus differentiates from the representative token-based AR model Chameleon (Team, 2024) only in the input embedding modules and output heads, we propose to substitute the VQ operation with a soft alternative and augment the model with an extra diffusion head to instantiate Orthus. We tune only the embedding modules and diffusion head (with 0.3B parameters in total) to reconstruct images on a 10k dataset to effortlessly obtain an Orthus base model. Orthus-base can further adopt post-training to bolster its ability to model interleaved images and text.

We have performed extensive studies to evaluate Orthus. For mixed-modality understanding and generation, Orthus outperforms the editing-specific model Instruct-pix2pix (Brooks et al., 2023) and exhibits in-context learning capabilities for unseen tasks. Furthermore, Orthus demonstrates a strong ability to generate logically coherent interleaved image-text content with high relevance. For visual understanding and generation, Orthus is substantially superior to Chameleon and Show-o (Xie et al., 2024) across multimodal understanding and generation tasks. Notably, Orthus achieves a GenEval (Ghosh et al., 2024) accuracy of 0.58 and a POPE score of 79.6, even surpassing specialized text-to-image models SDXL (Podell et al., 2023) and the performant InstructBLIP-13B (Dai et al., 2023).

To summarize, our contributions are as follows:

- We introduce Orthus for interleaved image-text generation. Orthus models the correlation between modalities through AR principle and generates discrete text tokens and continuous image features with dedicated heads.

- We propose an efficient strategy to build Orthus by exploiting its connection with existing unified AR models, which reduces the cost to merely 72 A100 GPU hours.

- Compared to related works such as Chameleon (Team, 2024) and Show-o (Xie et al., 2024), Orthus outperforms them across various visual understanding and generation benchmarks, while also demonstrating extra capabilities in mixed-modality understanding and generation, positioning it as a promising approach for unified multimodal modeling.

## 2. Related Work

**Visual understanding.** To enable multimodal large language models (MLLMs) to comprehend modalities beyond text, prior work has introduced methods that leverage pre-trained, modality-specific encoders (Radford et al., 2021; Li et al., 2022; Yu et al., 2022; Chen et al., 2024) to generate latent representations for each modality. These representations are then projected into a pre-trained LLM's input space through trained adapters, allowing for multimodal information alignment within the language model, understanding

are handled within the transformer backbone (Liu et al., 2024d; Zhu et al., 2023; Dai et al., 2023; Driess et al., 2023; Chen et al., 2023b; Liu et al., 2024b; Lin et al., 2024). This framework allows LLMs to perform complex multimodal tasks while maintaining the language-based reasoning capabilities inherent to their architecture.

**Visual generation.** The generation of visual content has long been a central focus within the deep learning research community (Kingma, 2013; Goodfellow et al., 2014; Karras et al., 2019; Vahdat & Kautz, 2020). Over the past few years, research in visual generation has focused on decomposing visual signals in a more sophisticated manner and generating them iteratively. Diffusion models (Sohl-Dickstein et al., 2015; Ho et al., 2020; Song et al., 2020b; Dhariwal & Nichol, 2021; Rombach et al., 2022; Peebles & Xie, 2023; Esser et al., 2024) transform generation into a reverse diffusion process from noise to data, gradually refining an initial noise input through a series of denoising steps. While another line of work aims to emulate the success of AR modeling from language modeling within the visual domain (Parmar et al., 2018; Razavi et al., 2019; Ramesh et al., 2021; Yu et al., 2023; Sun et al., 2024b). Specifically, images are first transformed into a sequence of vector-quantized tokens (Van Den Oord et al., 2017; Esser et al., 2021; Tian et al., 2024; Yu et al., 2024), after which AR modeling is then performed on the discrete-valued token space (Touvron et al., 2023a). To mitigate generation quality degradation caused by information loss during the VQ process, MAR replaces the per-token categorical distribution modeling with a diffusion procedure (Li et al., 2024; Fan et al., 2024). Our proposed method generalizes MAR to cross-modality generation.

**Unified visual understanding and generation.** To enable a model to possess both understanding and generation capabilities, one kind of approach aims to connect LLMs with multimodal adapters and diffusion decoders (Sun et al., 2023; Ge et al., 2024; Ye et al., 2024). However, using multiple distinct components can lead to modeling redundancy. Consequently, recent studies seek to leverage a single transformer for unified understanding and generation. A straightforward approach is to apply vector quantization to continuous visual signals to enable visual tokens, like discrete text tokens, to be trained within a unified token space using cross-entropy loss. Representative works are LWM (Liu et al., 2024e), Chameleon (Team, 2024), Anole (Chern et al., 2024), and VILA-U (Wu et al., 2024b). Alternatively, some works have explored combining AR with diffusion modeling. Show-o (Xie et al., 2024) unifies AR and discrete diffusion modeling for multimodal understanding and generation within one single transformer. Transfusion (Zhou et al., 2024) trains one shared transformer for both discrete text autoregression and continuous image diffusion. Our proposed method circumvents the potential information loss caused by quantization and noise disturbance.

## 3. Preliminary

Unified multimodal modeling aims to cope with a blend of images and texts with a single compact model (Liu et al., 2024f; Team, 2024; Wang et al., 2024; Zhou et al., 2024; Xie et al., 2024). The model usually includes a vision autoencoder, specified with an encoder $E$ and a decoder $D$, a text tokenizer, and a transformer network (Vaswani, 2017). The encoder $E$ is used to map the input image to a sequence of patch-wise features $V := [\boldsymbol{v}_1, \ldots, \boldsymbol{v}_n], \boldsymbol{v}_i \in \mathbb{R}^{d_v}$ for effective information compression, where $d_v$ is the feature dimension and $n$ is the number of patches. The text tokenizer maps the input text into a sequence of text tokens $U := [u_1, \ldots, u_m]$ with $m$ as the sequence length. The transformer is then asked to process $U$ and $V$ simultaneously to yield meaningful outputs, which can be then detokenized as texts or decoded by $D$ to produce images. There are primarily two strategies for the learning of the transformer, detailed as follows.

**Fully AR models.** Observing that the AR principle excels in the generative modeling of discrete content, seminal works, including LWM (Liu et al., 2024e) and Chameleon (Team, 2024), propose to leverage the Vector Quantization (VQ) (Van Den Oord et al., 2017) technique to transform the continuous image features $V$ as discrete tokens to enable a fully AR modeling of the mixture of images and texts. Specifically, VQ introduces a set of $K$ codes $\{\boldsymbol{c}_j \in \mathbb{R}^{d_v}\}_{j=1}^K$ and solves the following problem for continuous-to-discrete transformation:

$$\tilde{v}_i = \underset{j \in \{1, \ldots, K\}}{\arg\min} \ d(\boldsymbol{v}_i, \boldsymbol{c}_j) \text{ for } i = 1, \ldots, n, \quad (1)$$

where $d(\cdot, \cdot)$ is a distance metric.

Let $\tilde{V} := [\tilde{v}_1, \ldots, \tilde{v}_n]$ denote the discrete image tokens. The fully AR model embeds both $\tilde{V}$ and $U$ as $d_e$-dim features. Specifically, the embedding corresponding to $\tilde{v}_i$ is

$$\boldsymbol{h}_i = \sum_j \boldsymbol{w}_j \mathbb{1}_{\tilde{v}_i = j}, \quad (2)$$

where $\{\boldsymbol{w}_j \in \mathbb{R}^{d_e}\}_{j=1}^K$ refer to the embedding weights. The embeddings for text tokens can be similarly gained, yet with another set of embedding weights. The transformer then processes these embeddings with *causal attention*, where the output head naturally yields the prediction of the next token. For training, the objective is simply the AR loss.

Despite being simple, the fully AR models can suffer from information loss (Liu et al., 2024a; Team, 2024), because VQ makes the transformer unable to directly look at the image features $\boldsymbol{v}_i$.

**AR-diffusion mixed models.** Another line of unified multimodal models is AR-diffusion mixed models (Zhou et al., 2024; Xie et al., 2024; Zhao et al., 2024), which integrates

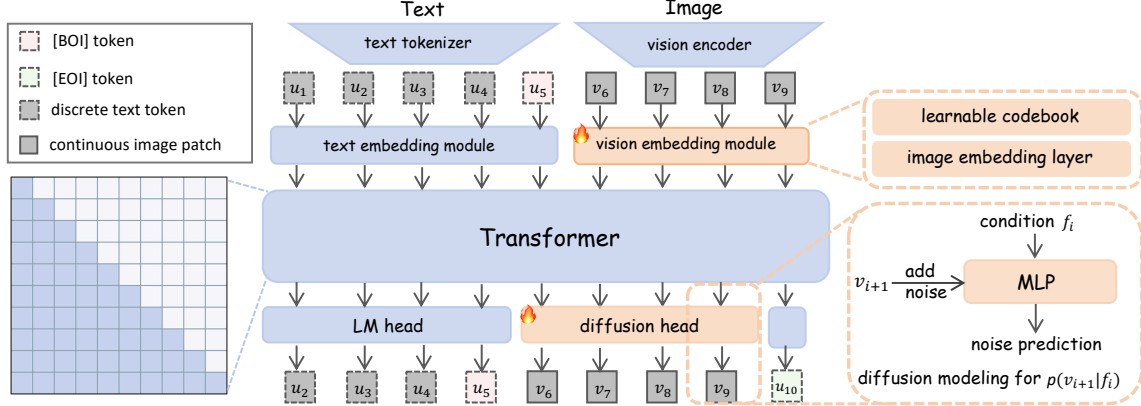

*Figure 2.* Architecture of Orthus. Orthus is composed of a text tokenizer, a vision autoencoder, two modality-specific embedding modules, a transformer backbone, and two modality-specific heads. Orthus tokenizes texts into discrete text tokens and encodes images into continuous patch-wise features. They are then embedded as a sequence of vectors and processed by the transformer backbone with causal attention, generating a sequence of output vectors. The vectors are routed to modality-specific heads, with the *LM head* to predict the next text token categorically and the *diffusion head* to predict the next image patch feature through conditional diffusion modeling.

diffusion modeling on images (Ho et al., 2020; Peebles & Xie, 2023) and AR modeling on text within a shared transformer. Take Transfusion (Zhou et al., 2024) for example, its inputs are a noisy version of the image features $V$, denoted as $\bar{V}$, and the text tokens $U$. To facilitate the simultaneous processing of $\bar{V}$ and $U$, the attention mask of the transformer adopts a unique configuration—with a full-attention structure among $\bar{V}$ and a causal structure among $U$. Then, the outputs from $\bar{V}$ are directed to an output projector to predict the noise on $\bar{V}$, whereas the outcomes linked to $U$ are channeled to an LM head for next-token prediction. The training objective is the combination of AR loss and denoising loss with a balancing factor. During inference, the model operates as an AR model to generate texts and as a diffusion model to craft images, with special tokens indicating mode switching.

However, diffusion modeling inherently requires feeding noisy inputs to the model, hindering joint modeling of visual understanding (requiring clean images) and generation (requiring noisy ones). For example, Transfusion identifies a nearly 15% performance drop in image captioning when full-range noise is introduced during training.

## 4. Method

We introduce Orthus to address the issues of existing works. This section begins with an overview of Orthus and then elaborates on an efficient training recipe for Orthus. We will also illustrate a post-training pipeline of Orthus.

### 4.1. Overview of Orthus

As shown in Figure 2, Orthus directly takes the continuous image features $V$ and discrete text tokens $U$ as input, which

avoids the pathologies caused by the quantized image features $\tilde{V}$ or noisy image features $\bar{V}$. $U$ and $V$ are embedded into the $d_e$-dim representation space with a differentiable vision embedding module (detailed in the next subsection) and the aforementioned discrete embedding module respectively. Subsequently, the embeddings are fed into the transformer backbone with *causal attention* for the modeling of both inter- and intra-modality interdependence. Given the output states of such a backbone contain enough information about the multimodal context, Orthus sends them to two modality-specific heads—a diffusion head and an LM head—to predict the next image patch or the next token.

Specifically, let $\boldsymbol{f}_i$ denote the output state corresponding to the input image feature $\boldsymbol{v}_i$ and $\epsilon_\theta$ denote the diffusion head employed by Orthus with parameter $\theta$. The goal of $\epsilon_\theta$ is to predict for the next patch feature $\boldsymbol{v}_{i+1}$ conditioning on $\boldsymbol{f}_i$. According to common practice (Ho et al., 2020; Dhariwal & Nichol, 2021), the learning objective for the diffusion head can be formalized as:

$$\mathcal{L}_{\text{diff}} = \mathbb{E}_{\boldsymbol{\epsilon},t}[\|\boldsymbol{\epsilon} - \epsilon_\theta(\sqrt{\overline{\alpha}_t}\boldsymbol{v}_{i+1} + \sqrt{1 - \overline{\alpha}_t}\boldsymbol{\epsilon}, t, \boldsymbol{f}_i)\|_2^2], \quad (3)$$

where $\boldsymbol{\epsilon} \sim \mathcal{N}(\boldsymbol{0}, \mathbf{I})$ is a Gaussian noise and $t$ is a randomly sampled timestep. $\overline{\alpha}_t$ follows a pre-defined noise schedule (Ho et al., 2020). In practice, $\epsilon_\theta$ can be a shallow multilayer perception (MLP) with three inputs (the condition $\boldsymbol{f}_i$, the scalar timestep $t$, and the noisy state). On the other hand, the LM head remains the compact linear projection followed by a softmax transformation to yield the predictive probability of the next token over the entire vocabulary.

### 4.2. An Efficient Strategy for Constructing Orthus-base

The differences between Orthus and fully AR models exist in the vision embedding module and the output head. Given

that pre-training a multimodal model from scratch can be frustratingly costly but the fully AR models like LWM (Liu et al., 2024e) and Chameleon (Team, 2024) are readily accessible from the open-source community, we are naturally interested in deriving Orthus based on them at a minimal expense. This section elaborates on a hard-to-soft adaptation trick and an efficient training strategy to enable this.

**Differentiable vision embedding module.** It is easy to note that the embedding yielded by Equations 1 and 2 can be equivalently obtained via a softmax-based transformation

$$h_i = \sum_j w_j \frac{e^{-d(v_i, c_j)/\tau}}{\sum_{k=1}^{K} e^{-d(v_i, c_k)/\tau}}, \qquad (4)$$

with $\tau \to 0$. Increasing $\tau$ gradually from 0 then naturally lifts the information bottleneck from the image features $v_i$ to the model outputs $f_i$, while rendering the reuse of the pre-trained weights and codes of fully AR models possible. This way, the codes $\{c_j\}_{j=1}^{K}$ also become a part of the input module, so we can leverage gradients to directly push them to adapt to the multimodal learning tasks. This contradicts fully AR models which froze the codes during training.

**Training strategy.** With the above trick, we start with a pre-trained fully AR model, transform its input module into a differentiable one, and introduce an output diffusion head to initialize Orthus. These modifications primarily focus on the visual part, thus we recommend fine-tuning the initialized model on a collection of images. In particular, we input only the image into Orthus to acquire the hidden states $f_i$ and utilize the diffusion loss in Equation 3 to recover the next patch to train the vision embedding module and diffusion head. The temperature $\tau$ is set to 1 during training.

Initialized with the typical Chameleon-7B (Team, 2024), Orthus can acquire image processing capabilities while preserving the text generation capacity after 9-hour training on 10k high-quality images (laion-coco aesthetic) using 8 A100 GPUs. We designate this model as Orthus-base, a pre-trained model capable of generating continuous image features and discrete text tokens.

Although the decoder in the VQ-VAE (Van Den Oord et al., 2017) used by Chameleon can reconstruct the raw image pixels given the patch-wise features $V$ to some extent, it can be suboptimal due to the quantization-aware training. To address this, we advocate further tuning its decoder to reconstruct high-quality images directly based on $V$. The comparison between the capacity of the original VQ-VAE and ours is exhibited in Appendix A.

### 4.3. Multimodal Post-training

Orthus-base can be further post-trained to unlock its potential for interleaved image-text modeling in complex downstream tasks. These include generating text (*e.g.*, visual question answering), images (*e.g.*, image editing), or even both images and text (*e.g.*, storybook generation) from mixed-modality inputs. For example, given $[V, U]$ as user input and $[V, U, V, U]$ as model output, we surround image features $V$ with the embeddings of special tokens [BOI] and [EOI] before the concatenation with $U$. A [SEP] token is used to separate user input and model output in each conversation.

Let $\mathcal{L}_{ar}$ denote the AR loss on the text tokens. The entire training objective of Orthus is then $\mathcal{L}_{Orthus} = \mathcal{L}_{ar} + \lambda \mathcal{L}_{diff}$, where $\lambda$ is a balancing coefficient. All parameters except for those of the vision autoencoder are tuned. Hereinafter, we will denote the model trained following this objective as Orthus, distinguishing it from Orthus-base.

During inference, Orthus alternates between *next-token prediction* and *next-patch prediction* to seamlessly generate interleaved texts and images. When [BOI] is sampled during the next-token prediction process, the algorithm moves to *next-patch prediction*. Once a fixed number of $n$ image patches are generated, [EOI] is appended and the algorithm switches back to *next-token prediction*.

## 5. Experiments

In this section, we evaluate Orthus's performance in interleaved image-text modeling as well as visual understanding and generation. Both quantitative and qualitative results demonstrate the effectiveness of Orthus.

### 5.1. Implementation Details

We implement the diffusion head as an MLP consisting of 3 residual blocks, each sequentially applying AdaLN (Peebles & Xie, 2023), a linear layer (width of 1536 channels), SiLU activation, and another linear layer. The condition vector $f_i$ is added to the diffusion time embedding, which is then incorporated through AdaLN. The diffusion noise schedule is linear following (Rombach et al., 2022), with 1000 steps at training time. $\lambda$ is set to 100 to balance the order of magnitude between $\mathcal{L}_{diff}$ and $\mathcal{L}_{ar}$ during post-training. During inference, we use greedy decoding to generate text. For image generation, we adopt the DDIM (Song et al., 2020a) sampler with 100 steps. We employ classifier-free guidance (CFG) (Ho & Salimans, 2022) with the scale set to 5 during sampling. All images are generated at a resolution of 512×512. More training and evaluation details are provided in Appendix B.

### 5.2. Interleaved Image-Text Generation

Compared to existing unified models, such as Janus-series (Wu et al., 2024a; Ma et al., 2024), that focus exclusively on visual understanding and generation, we inves-

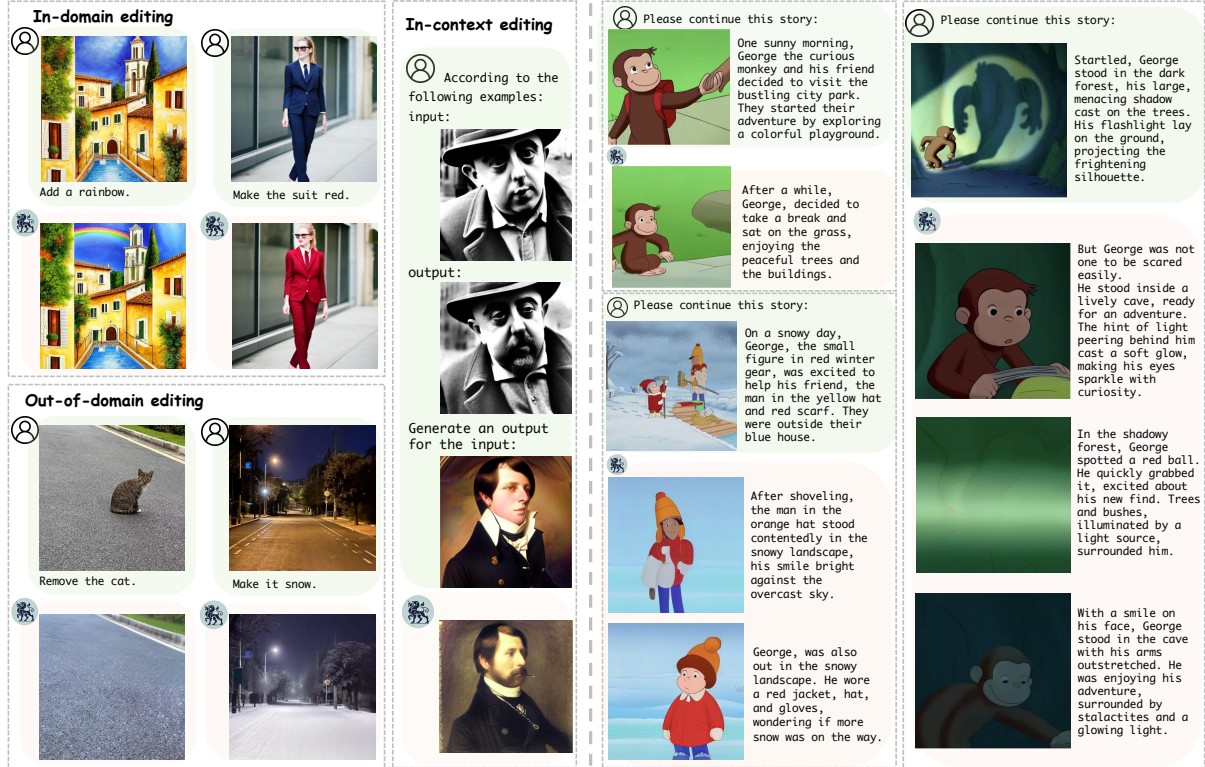

*Figure 3.* Qualitative results on mixed image-text understanding and generation of Orthus. **Left:** Image editing results after fine-tuned on Instruct-Pix2Pix (Brooks et al., 2023). Notably, Orthus exhibits *in-context learning* capacity by performing image editing successfully when provided with examples rather than explicit instructions, which is not included in the training dataset. **Right:** Interleaved storybook creation results after finetuned on the StoryStream (Yang et al., 2024b) dataset. Results show that Orthus excels in generating logically coherent interleaved image-text with high relevance.

*Table 1.* Comparisons of CLIP similarities (Ruiz et al., 2023; Gal et al., 2022) between editing-specific diffusion models and Orthus on the test dataset of Instruct-Pix2Pix.

| Model | -T↑ | -I↑ | -D↑ |
|---|---|---|---|
| PnP (Tumanyan et al., 2023) | 0.156 | 0.76 | 0.023 |
| SDEdit (Meng et al.) | 0.229 | 0.84 | 0.047 |
| I-Pix2Pix (Brooks et al., 2023) | 0.233 | **0.88** | 0.045 |
| **Orthus (Ours)** | **0.238** | 0.87 | **0.049** |

tigate Orthus's flexibility and extensibility to model interleaved images and text on two representative downstream tasks: image editing and storybook generation.

**Image-Text → Image.** We compare the performance of Orthus with an editing-specific diffusion model after training Orthus-base on the 400k Instruct-pix2pix (Brooks et al., 2023) training dataset. Table 1 shows that the images edited by Orthus align well with both the given instruction and the input image, performing comparably to or even surpassing the editing-specific diffusion model. Moreover, Figure 3 illustrates Orthus's strong generalization ability to edit images in zero-shot real-image domains. Notably, Orthus exhibits *in-context learning capacity* as a unified multimodal model:

when provided with examples instead of explicit instructions that do not match the formats seen during training, it successfully completes the task. This highlights Orthus's strong capability for interleaved data modeling and its great potential as a foundation multimodal model. More editing examples and comparisons are provided in Appendix F.

**Image-Text → Image-Text-Image-Text.** To further validate Orthus's superiority in modeling interleaved data, we fine-tune Orthus-base with the unified learning objective on the StoryStream (Yang et al., 2024b) dataset, which includes a collection of images and corresponding narratives from cartoon series. As shown in Figure 3, after training, Orthus can generate contextually consistent scenes paired with narrative text given an initial image-text pair and the instruction "Please continue this story." Notably, there is a strong alignment between images and text (e.g., the smile on the monkey's face) as well as consistent detail across images (e.g., the boy's orange hat and red scarf). These results highlight Orthus's ability to generate long sequences of contextually relevant images and text. This capability opens up potential applications such as report generation, educational content creation, and other tasks requiring seamless mixed-modality content generation.

*Table 2.* **Evaluation on visual understanding benchmarks.** Und. and Gen. denote "understanding" and "generation", respectively. Models using external pre-trained diffusion models are marked with * and Chameleon[†] is post-trained with the same dataset as Orthus. The results in **bold** and underline are the best and second-best results, respectively. The results correspond to the exact match accuracy.

| Type | Model | # Params | POPE↑ | MME-P↑ | VQAv2↑ | GQA↑ | MMMU↑ |
|------|-------|----------|-------|--------|--------|------|-------|
| Und. Only | LlaVa (Liu et al., 2024d) | 7B | 76.3 | 809.6 | - | - | - |
| | LlaVA-v1.5 (Liu et al., 2024b) | 7B | 85.9 | 1510.7 | 78.5 | 62.0 | 35.4 |
| | InstructBLIP (Dai et al., 2023) | 7B | - | - | - | 49.2 | - |
| | Qwen-VL-Chat (Bai et al., 2023) | 7B | - | 1487.5 | 78.2 | 57.5 | - |
| | Emu3-Chat (Wang et al., 2024) | 8B | 85.2 | 1243.8 | 75.1 | 60.3 | 31.6 |
| | InstructBLIP (Dai et al., 2023) | 13B | 78.9 | 1212.8 | - | 49.5 | - |
| Und. and Gen. | Emu* (Sun et al., 2023) | 13B | - | - | 52.0 | - | - |
| | NExT-GPT* (Wu et al., 2013) | 13B | - | - | **66.7** | - | - |
| | Gemini-Nano-1 (Team et al., 2023) | 1.8B | - | - | 62.7 | - | 26.3 |
| | Show-o (Xie et al., 2024) | 1.3B | 73.8 | 948.4 | 59.3 | 48.7 | 25.1 |
| | LWM (Liu et al., 2024e) | 7B | 75.2 | - | 55.8 | 44.8 | - |
| | Chameleon[†] | 7B | 77.8 | 1056.9 | 57.8 | 49.6 | 26.7 |
| | **Orthus (Ours)** | 7B | **79.6** | **1265.8** | 63.2 | **52.8** | **28.2** |

## 5.3. Visual Understanding and Generation

In this section, we validate the effectiveness of Orthus on visual understanding and generation by post-training Orthus-base with a mixture of LlaVA-v1.5-665K (Liu et al., 2024d) and high-quality text-to-image data (JourneyDB (Sun et al., 2024a) and LAION-COCO-aesthetic (laion-coco aesthetic) recaptioned from ShareGPT-4v (Chen et al., 2023a)). We also fine-tune pre-trained Chameleon (Chern et al., 2024) with the same mixed dataset as Orthus to provide an apple-to-apple baseline.

**Image → Text.** Table 2 shows that: (i) Compared to Chameleon post-trained with the same dataset, Orthus consistently demonstrates superior performance across all benchmarks. Besides, inspecting OCR-related tasks in MME-P, we witness a significant superiority of Orthus over Chameleon (with scores of 70 vs. 45). These results validate the superiority of Orthus's modeling by adopting lossless representations for images. (ii) Orthus outperforms other unified models using a single transformer like LWM and Show-o across all benchmarks, highlighting its efficacy for unified modeling. (iii) Compared to larger unified models using an external diffusion model, such as NExT-GPT-13B, Orthus achieves decent results on the VQAv2 benchmark. It is reasonable to speculate that Orthus's potential for multimodal understanding problems can be further unleashed by scaling up training compute and data.

**Text → image.** Table 3 shows that: (i) When compared with strong competitors specialized for text-to-image generations such as DALL-E 2 and SDXL, Orthus achieves an improvement of 0.06 and 0.03 on GenEval, respectively. (ii) Compared to Chameleon and its post-trained version, Orthus demonstrates significant superiority on both GenEval and

*Table 3.* **Comparison with state-of-the-arts on visual generation benchmarks.** Model using external pre-trained diffusion model is marked with * and Chameleon[†] is post-trained with the same dataset as Orthus. The results in **bold** and underline are the best and second-best results, respectively.

| Type | Model | Res. | GenEval | HPS |
|------|-------|------|---------|-----|
| Gen. Only | SDv1.5 (Rombach et al., 2022) | 512 | 0.43 | 27.0 |
| | SDv2.1 (Rombach et al., 2022) | 512 | 0.50 | 27.2 |
| | DALL-E (Ramesh et al., 2022) | 512 | 0.52 | 26.9 |
| | Emu3-Gen (Wang et al., 2024) | 512 | 0.54 | - |
| | SDXL (Podell et al., 2023) | 512 | 0.55 | 30.9 |
| | SD3(d=30) (Esser et al., 2024) | 512 | 0.64 | - |
| Und. & Gen. | SEED-X* (Ge et al., 2024) | 448 | 0.49 | - |
| | LWM (Liu et al., 2024e) | 256 | 0.47 | 26.1 |
| | Show-o (Xie et al., 2024) | 256 | 0.53 | 27.3 |
| | Transfusion (Zhou et al., 2024) | 256 | **0.63** | - |
| | Chameleon[†] | 512 | 0.43 | 26.9 |
| | **Orthus (Ours)** | 512 | 0.58 | **28.2** |

HPSv2. This advantage can be attributed to the utilization of continuous image representations and diffusion-based continuous modeling, which facilitates the generation of high-quality images with richer detail and stronger alignment with human preferences. (iii) Compared with other unified models such as SEED-X, LWM, and Show-o, Orthus obtains significantly better performance, highlighting the advantages of its modeling strategy. (iv) Qualitative results in Figure 4 showcases images generated by Orthus alongside results from other unified models, including Chameleon and Show-o. Results show that Orthus is capable of generating diverse, engaging, and realistic visual imagery at the resolution of 512×512.

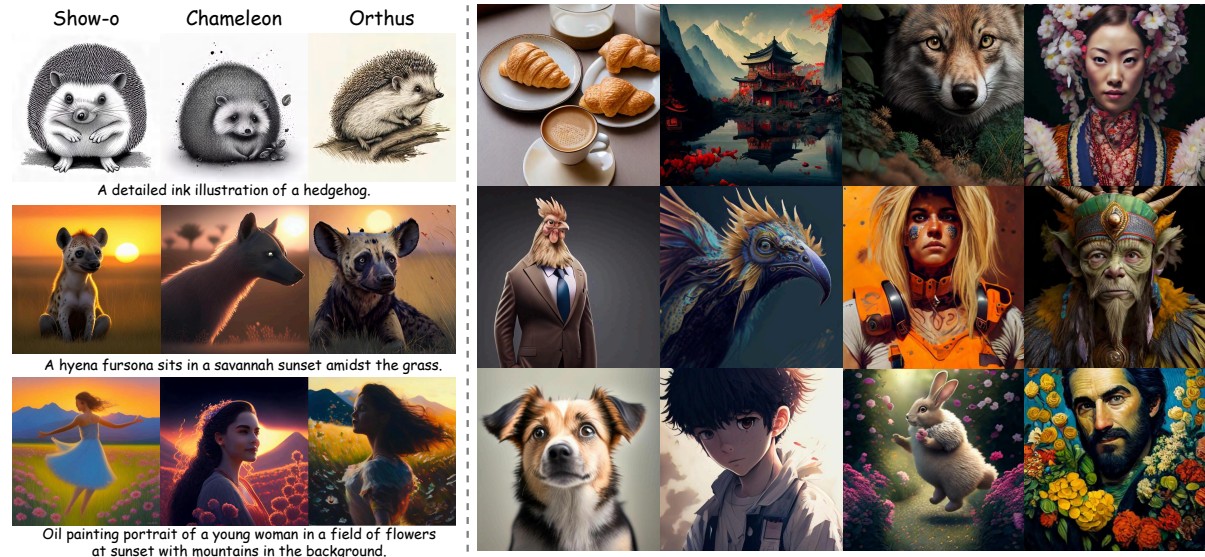

*Figure 4.* **Left:** Comparison between images generated by Show-o, Chameleon, and Orthus based on the same prompts. Samples produced by Orthus contain more details. **Right:** Text-to-image gallery of Orthus.

*Table 4.* Comparisons of the performance of Orthus via separate training and unified training across multimodal benchmarks.

| Type | $\mathcal{L}_{\text{diff}}$ | $\mathcal{L}_{\text{ar}}$ | POPE↑ | MME-P↑ | GQA↑ | GenEval↑ |
|---|---|---|---|---|---|---|
| Und. only | ✗ | ✓ | 78.7 | 1244.2 | 51.9 | - |
| Gen. only | ✓ | ✗ | - | - | - | 0.56 |
| Und. & Gen. | ✓ | ✓ | **79.6** | **1265.8** | **52.8** | **0.58** |

*Table 5.* Ablation study on the choice of vision embedding modules on visual understanding tasks.

| Type | POPE↑ | MME-P↑ | VQAv2↑ | GQA↑ | MMMU↑ |
|---|---|---|---|---|---|
| softmax | **78.7** | **1244.2** | **60.8** | **51.9** | **28.0** |
| argmin | 77.6 | 1064.8 | 57.9 | 50.1 | 26.7 |
| linear | 70.4 | 800.7 | 50.3 | 44.5 | 22.3 |

## 5.4. Ablation Studies

**Separate training vs. unified training.** To validate the efficacy of Orthus for unified multimodal modeling, we compare baselines using identical training data but with different learning objectives: (i) a generation-only baseline focused solely on text-to-image generation; (ii) an understanding-only baseline dedicated to visual understanding tasks; and (iii) a unified training objective, the default setting in Orthus. Table 4 shows that unified training outperforms task-specific training in both visual understanding and generation, highlighting the superiority of Orthus which facilitates information gains from bidirectional cross-modal learning.

**Impact of vision embedding modules on visual understanding tasks.** In this section, we ablate the impact of different choices of vision embedding modules to build Orthus from fully AR models on visual understanding. When we retain the original embedding module in fully AR models ("argmin" in Table 5), a performance drop is observed due to the information loss. Moreover, replacing the embedding module with a randomly initialized linear layer also leads to suboptimal performance due to the significant distribution shift between the embedded space and the transformer's input space. This misalignment may necessitate training with more image-text pairs to mitigate.

**Loss design.** To test the necessity of diffusion modeling for the image features, we train the MLP head with straightforward Mean Squared Error (MSE) loss between predictions and target features. As shown in Appendix C, the model trained with MSE loss generates degraded samples that lack details and exhibit limited color diversity. The reason is that the deterministic nature of MSE loss leads to mode collapse.

## 6. Conclusion

In this paper, we propose Orthus, a unified multimodal model for interleaved image-text understanding and generation. Orthus generates content across modalities by routing outputs from a shared transformer backbone to modality-specific heads. Its continuous treatment of visual signals preserves input integrity and its unified AR modeling approach for both discrete text tokens and continuous image features enables superior performance across various multimodal understanding and generation benchmarks. For future work, we plan to scale Orthus by increasing its parameters and leveraging larger, interleaved datasets to maximize its potential. We also aim to broaden its capabilities by incorporating more modalities, including video and audio.

**Limitations.** The main limitations of this work are as follows. Orthus exhibits relatively high inference latency intro-

duced by the need for multiple forward passes through the diffusion head. Besides, Orthus is limited to 7B parameters due to constrained computational resources.

## Impact Statement

This work presents a challenge in machine learning and proposes a solution, the potential negative consequences are not apparent. While it is theoretically possible for any technique to be misused, the likelihood of such misuse occurring at the current stage is low.

## Acknowledgements

This work was supported by NSF of China (Nos. 92470118, 62306176), Natural Science Foundation of Shanghai (No. 23ZR1428700), CCF-Zhipu Large Model Innovation Fund (No. CCF-Zhipu202412), and CCF-ALIMAMA TECH Kangaroo Fund (NO. CCF-ALIMAMA OF 2025010).

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

## A. Comparison of Vision Autoencoder

To construct a vision autoencoder capable of decoding high-quality images based on continuous image features $V$, we freeze the encoder of Chameleon's VQ-VAE, drop the quantization step, and finetune the decoder only to reconstruct images, transforming it into a conventional continuous autoencoder effectively. The decoder is trained on LAION-Aesthetic dataset (laion-coco aesthetic)using a learning rate of 1e-5, a batch size of 256, and a total of 15,000 training steps. Table 6 shows that our vision autoencoder achieves better reconstruction quality compared to the original VQ-VAE. The evaluation is conducted on a subset of the LAION-Aesthetic, consisting of 10,000 images that are excluded from the training dataset.

*Table 6.* Comparison of reconstruction quality for vision autoencoders: the discrete one is worse than the continuous variant.

| Model | PSNR↑ | SSIM (Wang et al., 2004)↑ |
|---|---|---|
| VQ-VAE (Team, 2024) | 23.7 | 0.80 |
| Ours | 26.1 | 0.84 |

## B. Training details

The images for training Orthus-base are the first 10k from laion-coco aesthetic. Both training and evaluation are carried out on servers equipped with 8 NVIDIA A100 80GB GPUs.

*Table 7.* Training details for constructing Orthus-base and the instruction-tuned one for visual understanding and generation in 5.3.

| Model | Orthus-base | Instruct-tuning |
|---|---|---|
| Optimizer | AdamW ($\beta_1 = 0.9$, $\beta_2 = 0.99$) | |
| Learning Rate | 1e-4 | 1e-5 |
| Batch Size | 32 | 16 |
| Training Steps | 15,000 | 35,000 |

## C. Diffusion Loss v.s. MSE Loss

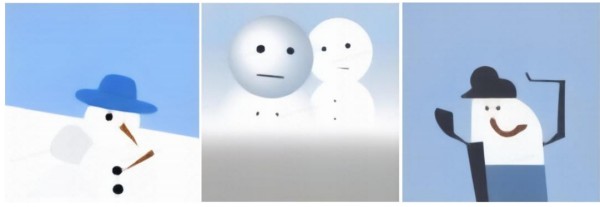

*Figure 5.* Text-to-image results from models trained with MSE loss. The text prompt is "Generate an image of a snowman."

## D. Examples on Visual Generation

Figure 6 shows examples of images generated from Orthus post-trained in Section 5.3.

## E. Examples on Visual Understanding

In addition to quantitatively evaluating Orthus in Section 5.3 on domain-specific tasks, we also assess its performance in general chat scenarios in Figure 7.

## F. Examples on Image Editing

Figure 8 shows random examples of image editing by Orthus-base post-trained on Instruct-Pix2Pix (Brooks et al., 2023).

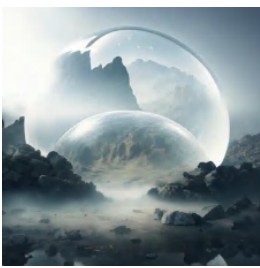

A photorealistic image of a giant floating glass sphere in a rocky landscape surrounded by a gentle mist.

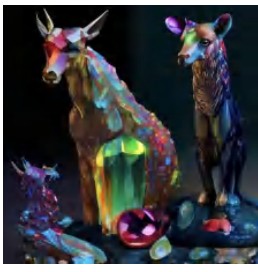

Animals fashioned from gems, colorful and shapely, depicted in natural lighting, with a slight effervescence.

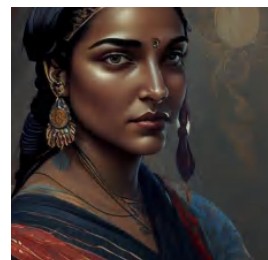

A portrait painting of a South Indian woman wearing a sari with intricate details and an eerie sense of horror, created in ultra-realistic style by artgerm.

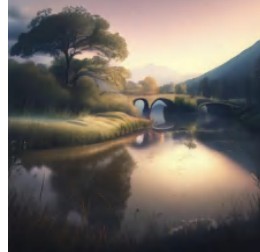

A serene meadow with a tree, river, bridge, and mountains in the background under a slightly overcast sunrise sky.

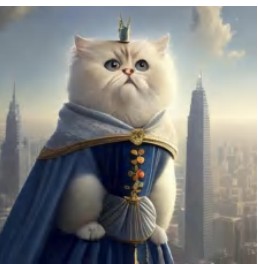

A painting of a Persian cat dressed as a Renaissance king, standing around a skyscraper overlooking a city.

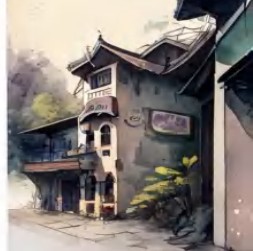

A stylized building in watercolor gouache, featuring interesting shapes and forms, located in a desolate landscape with a food stall in an Asian-style alleyway.

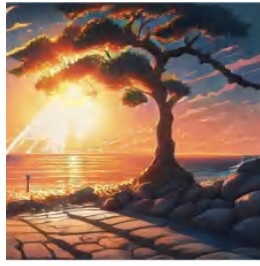

A cobblestone street with a tree over the sea at sunset, illuminated by sun rays, in a colorful illustration by Peter Chan on Artstation.

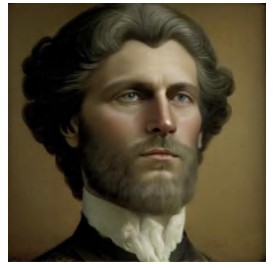

Portrait of Archduke Franz Ferdinand by Charlotte Grimm, depicting his detailed face.

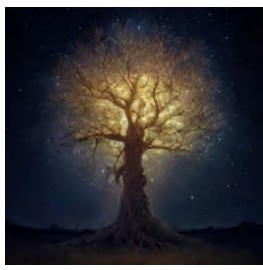

A glowing dry tree stands alone under a starry sky in a detailed fantasy artwork

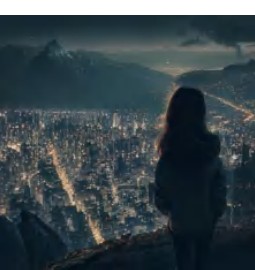

A girl looks out from the edge of a mountain onto a large city at night.

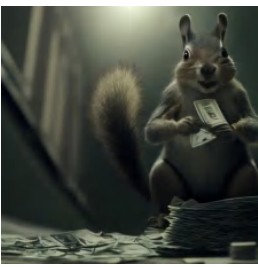

A gangster squirrel is counting his money in a low angle film still.

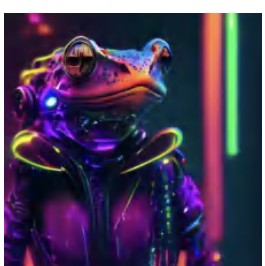

A neon-colored frog in a cyberpunk setting.

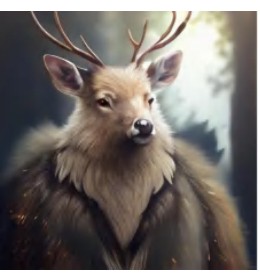

Digital painting of a furry deer character on FurAffinity.

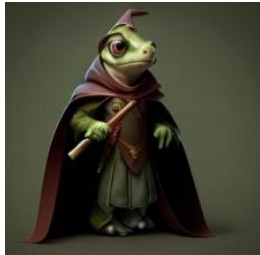

An anthropomorphic frog wizard wearing a cape and holding a wand.

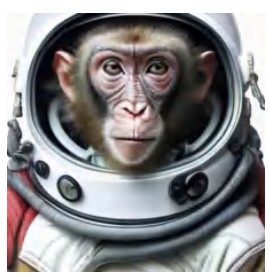

Portrait of a monkey wearing an astronaut helmet.

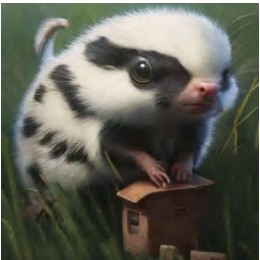

A close-up oil painting of a littlest pet shop fuzzy skunk in a field.

*Figure 6.* Generated $512 \times 512$ images from Orthus. Results demonstrate its ability to generate diverse, engaging, and realistic images.

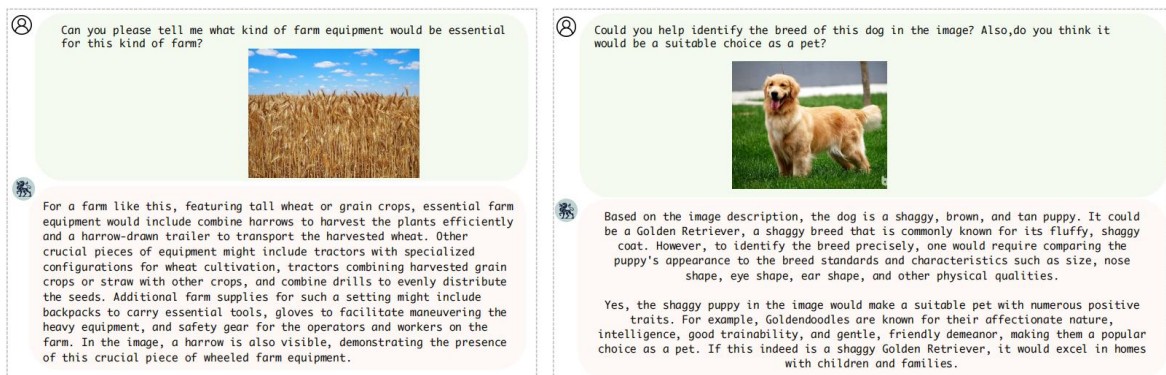

*Figure 7.* Examples of Orthus on visual understanding. Results demonstrate that Orthus exhibits strong instruction-following capabilities and robust generalization abilities.

| | Have the animals be made of chocolate | Add a lightning storm | Change the house to a castle | Make it more apocalyptic |
|---|---|---|---|---|

*Figure 8.* Examples of Orthus on image editing. Compared to editing-specific diffusion models ([Brooks et al., 2023](#)), Orthus demonstrates better fidelity to the original image in regions where no editing is required.

