# OpenReview forum: "Orthus: Autoregressive Interleaved Image-Text Generation with Modality-Specific Heads"
_ICML.cc/2025/Conference — ICML 2025 poster_

### Official Review · Reviewer_GAez · 2025-02-15

**Overall Recommendation:** 3

**Summary:**

This paper introduce Orthus, which is  a unified multimodal LLM for generating interleaved images and
text from mixed-modality inputs by simultaneously handling discrete text tokens and continuous image features under the AR modeling
principle.

## update after rebuttal
After considering the authors' response, I am inclined to maintain my original rating of 'Weak Accept.' This decision is based on the model's capability for mixed image-text understanding and generation, but relatively weak vision embedding module.

**Claims And Evidence:**

The claims made in the submission are supported by clear and convincing evidence.

**Essential References Not Discussed:**

The authors discussed sufficient related works.

**Experimental Designs Or Analyses:**

The experimental designs and analyses are reasonable to me.

**Methods And Evaluation Criteria:**

The methods and evaluation criteria make sense for the unified multimodal LLM .

**Other Comments Or Suggestions:**

No.

**Other Strengths And Weaknesses:**

Strengths:
1. The paper is generally well-written and easy to follow, with clearly illustrated figures.
2. Orthus is capable of  mixed image-text understanding and generation such as in-context editing and interleaved storybook
creation.

Weaknesses:
1. Orthus does not perform exceptionally well in multimodal comprehension, which may be limited by its relatively weak vision embedding module.
2. The authors build Orthus based on Chameleon and substitute the VQ operation with a soft alternative and only tune the parameters of vision embedding module and diffusion head. Have the author tried using a pre-trained multimodal LLM based on a continuous ViT representation, with an added diffusion head to achieve unified multimodal understanding and generation?

**Questions For Authors:**

Why does unified training perform better than understanding training only on the multimodal comprehension benchmark as listed in Tab 4? This seems rarely seen in unified multimodal understanding and generation work. For example, the Emu series conducts separate instruction tuning for understanding and generation.

**Relation To Broader Scientific Literature:**

LLM-based unified multimodal comprehension and generation is a very important research direction, especially for interleaved image-text generation given mixed-modality input. This approach is quite inspiring as it overcomes the limitations of diffusion modeling by equipping a multimodal LLM with an additional diffusion head.

**Theoretical Claims:**

No proofs for theoretical claims in this paper.

---

> ### Author Rebuttal · Authors · 2025-03-31
>
> Thank you for your attentive comments! We are glad you thought our paper was well-written and easy to follow. We address your  concerns point by point below.
>
> **W1:** Orthus does not perform exceptionally well in multimodal comprehension, which may be limited by its relatively weak vision embedding module.
>
> Yes. The features encoded by VAE are suited for visual generation, but they can be suboptimal for multimodal comprehension compared to CLIP-ViT due to the lack of semantic information. However, in this paper, we focus on building a unified model capable of interleaved image-text understanding and generation, where images need to be understood and generated simultaneously. Therefore, we embed images using VAE instead of CLIP-ViT. **A possible improvement is to incorporate semantic loss during the training of the VAE.** Additionally, we train solely on the LlaVa-v1.5-mix-665k dataset for multimodal comprehension currently, demonstrating the advantages of the improved soft vision embedding module and the continuous treatment of visual signals. In the future, we plan to incorporate more high-quality visual understanding data for further improvement.
>
> **W2:** The authors build Orthus based on Chameleon and substitute the VQ operation with a soft alternative and only tune the parameters of vision embedding module and diffusion head. Have the author tried using a pre-trained multimodal LLM based on a continuous ViT representation, with an added diffusion head to achieve unified multimodal understanding and generation?
>
> Thank you for this interesting suggestion. It is indeed possible to train an additional diffusion head to predict representations encoded by CLIP-ViT autoregressively. However, **decoding features encoded by CLIP-ViT back into the original image is challenging.** Previous work like VQKD[1] has attempted this but the reconstructed images suffered from significant blurring and an evident loss of high-frequency details, which is attributed to the emphasis on encoding semantic information at the expense of fine-grained details.
>
> Alternatively, prior studies [2] have successfully reconstructed images using a CLIP encoder and a diffusion-based DiT decoder. However, **this design introduces redundancy by modeling the correlation between image patches both in the transformer backbone and diffusion decoder**, and the inclusion of two separate diffusion processes significantly slows down image generation.
>
> [1] Beit v2: Masked image modeling with vector-quantized visual tokenizers, arXiv:2208.06366.
>
> [2] 4M-21: An Any-to-Any Vision Model for Tens of Tasks and Modalities, NIPS 2024
>
> **Question:** Why does unified training perform better than understanding training only on the multimodal comprehension benchmark as listed in Tab 4? This seems rarely seen in unified multimodal understanding and generation work. For example, the Emu series conducts separate instruction tuning for understanding and generation.
>
> The synergy may stem from unified cross-modal learning, which renders the characterization of the correlation between modalities. Moreover, both generation and understanding tasks contribute to the alignment of text and image in a shared representation space. On the other hand, previous work DreamLLM [1] also observes a synergy between generation and understanding. It suggests that training LLMs with stronger visual comprehension capabilities leads to more fine-grained encoding of text, which in turn improves text-to-image tasks.
>
> [1] DreamLLM: Synergistic Multimodal Comprehension and Creation, ICLR 2024.
>
> We hope these reclarifications and explanations of our method in response to your initial concerns can convince you to increase your score.

---

### Official Review · Reviewer_ofPN · 2025-03-13

**Overall Recommendation:** 4

**Summary:**

The paper introduces Orthus, a unified multimodal model that autoregressively generates interleaved images and text. The key idea is to handle discrete text tokens and continuous image features within a single transformer framework by employing modality-specific heads. One head is dedicated to language modeling for predicting text tokens, while the other is a novel diffusion head designed to generate continuous image patches. This design avoids the information loss typically associated with vector quantization and mitigates the noise issues inherent in diffusion models when applied jointly with text. An efficient training strategy is proposed where a pre-trained autoregressive model is adapted by replacing hard vector quantization with a soft, differentiable alternative and adding a diffusion head and fine-tuning it on a modest image dataset. Experimental results demonstrate that Orthus outperforms existing unified models on benchmarks for both visual understanding and generation, including tasks like image editing and storybook creation, showing its potential for coherent interleaved image-text generation.

**Claims And Evidence:**

Overall, many claims are supported by extensive experiments and ablation studies. However, a few claims are still with some issues:

1. The paper states that using continuous image features avoids “information loss” and is entirely lossless. While benchmark improvements (e.g., higher MME-P and OCR scores) suggest benefits, the claim is potentially exaggerated since the diffusion process itself might introduce trade-offs that aren’t fully analyzed.

2. The claim that the Orthus-base model can be built “effortlessly” in only 72 A100 GPU hours is not contextualized with comparisons to similar models, making it hard to assess the real efficiency gain.

3. The demonstration of robust in-context learning for tasks such as image editing and storybook generation is mainly qualitative. More rigorous quantitative evaluation or statistical analysis would strengthen the evidence for these capabilities.

These points, if addressed, would make the paper’s claims even more convincing.

**Essential References Not Discussed:**

Please consider discussing (or even supplying experiments) the latest benchmarks, including MMIE and OpenING.

Xia, Peng, Siwei Han, Shi Qiu, Yiyang Zhou, Zhaoyang Wang, Wenhao Zheng, Zhaorun Chen et al. "Mmie: Massive multimodal interleaved comprehension benchmark for large vision-language models." arXiv preprint arXiv:2410.10139 (2024).
Zhou, Pengfei, Xiaopeng Peng, Jiajun Song, Chuanhao Li, Zhaopan Xu, Yue Yang, Ziyao Guo et al. "GATE OpenING: A Comprehensive Benchmark for Judging Open-ended Interleaved Image-Text Generation." arXiv preprint arXiv:2411.18499 (2024).

**Experimental Designs Or Analyses:**

The experimental design appears generally sound. The paper evaluates Orthus across multiple tasks, such as interleaved image-text generation, visual understanding (using benchmarks like VQAv2, GQA, and OCR-related metrics), and text-to-image generation, with comparisons against established baselines. Ablation studies, like those on the choice of vision embedding modules and the impact of unified versus separate training, also add depth to the analysis. However, there are a few points to note:

1. The evaluation of in-context learning and qualitative tasks (e.g., storybook generation) relies largely on qualitative observations with limited quantitative backing.

2. There is limited discussion on hyperparameter sensitivity and statistical significance of the reported improvements.

**Methods And Evaluation Criteria:**

The proposed methods are well-suited for the interleaved image-text generation problem. The use of modality-specific heads, which is a language modeling head for text tokens and a diffusion head for continuous image features, provides a clear and innovative way to address the shortcomings of vector quantization and noise issues in existing models. The evaluation criteria, including metrics like GenEval, MME-P, CLIP similarity scores, and OCR performance, are appropriate to assess both visual understanding and generation. However, additional quantitative analysis, especially regarding in-context learning, would further strengthen the evidence supporting these methods.

**Other Comments Or Suggestions:**

1. Some minor typographical and formatting issues could be addressed (e.g., consistent capitalization in headings and figure references).
2. The discussion on temperature scheduling for the softmax-based vision embedding could be expanded for clarity.
3. More detailed information on the dataset selection and preprocessing would improve reproducibility.
4. Some key parts are not explained. E.g., d=30.

**Other Strengths And Weaknesses:**

Strengths:
1. The paper presents an innovative integration of autoregressive modeling with modality-specific heads, effectively combining continuous image representation and discrete text generation.
2. It offers a practical and efficient training strategy by adapting pre-trained AR models, which could lower the barrier for building unified multimodal models.
3. Extensive experiments, including ablation studies and diverse benchmarks, provide solid empirical support for the approach.
4. The method clearly advances interleaved image-text generation, a challenging yet important task in multimodal learning.

Weaknesses:
1. Some claims, such as achieving “lossless” image representation, may be overstated given that the diffusion process could introduce its own trade-offs.
2. The efficiency claim (e.g., 72 A100 GPU hours) lacks sufficient comparative context with similar models, making it hard to gauge its significance.
3. Evaluation of in-context learning and qualitative tasks relies heavily on visual and anecdotal evidence, with limited rigorous quantitative analysis.
4. The description of certain architectural details and training trade-offs could be more explicit, particularly regarding hyperparameter sensitivity and noise management in the diffusion head.

**Questions For Authors:**

1. Could you provide more context or comparative baselines regarding the claim of 72 A100 GPU hours? How does this compare to similar models trained from scratch?
2. What is the rationale behind the specific temperature schedule used in the softmax-based vision embedding module, and how sensitive is the model performance to variations in this hyperparameter?
3. Can you offer additional quantitative evaluations to support the claims of in-context learning, particularly for tasks like storybook generation?
4. How does the model handle imbalanced or partially missing interleaved inputs, and have you evaluated its robustness in such scenarios?
5. Are there any failure cases or limitations observed during experiments that were not discussed in the paper? How might these impact the model's application in real-world tasks?

**Relation To Broader Scientific Literature:**

The paper’s contributions build directly on a range of well-established ideas in multimodal learning. First, it extends the vector quantization‐based autoregressive approaches (e.g., VQ-VAE methods as in Van den Oord et al., 2017) by replacing the hard quantization step with a soft, differentiable alternative. This adjustment seeks to mitigate the known information loss issues of converting continuous image features to discrete tokens. Second, by incorporating a dedicated diffusion head for generating continuous image patches, the paper leverages advancements in diffusion models (as seen in Ho et al., 2020 and Dhariwal & Nichol, 2021), which have proven effective in high-fidelity image synthesis.

Moreover, the work positions itself against recent unified AR-diffusion models (like Transfusion and Monoformer) by decoupling the noise-inducing aspects of diffusion from the core transformer backbone. This contrasts with approaches that mix noisy image inputs with text tokens, which can hurt performance in tasks such as image captioning. The paper also parallels the masked autoregressive (MAR) models but argues that its fully autoregressive formulation better captures interleaved image-text dependencies without complex hyperparameter tuning.

Finally, the efficient adaptation of pre-trained AR models—requiring relatively low training compute—aligns with current trends in making large-scale multimodal models more accessible and versatile. Overall, the paper synthesizes and extends prior findings in both language modeling and image generation, integrating them into a unified framework for interleaved image-text processing.

**Theoretical Claims:**

The paper does not offer new, formal theoretical proofs but builds on established formulations. For example, the derivation of the softmax-based alternative to vision embedding (Equation 4) is consistent with the standard property that a softmax with a temperature approaching zero approximates an argmax operation, as used in Equation 1. Similarly, the diffusion loss (Equation 3) follows common practice in diffusion models. No rigorous proofs regarding convergence or guarantees of “losslessness” are provided; instead, these claims are supported by empirical evidence. Overall, the standard derivations appear correct, though the paper could benefit from a more formal discussion of any theoretical guarantees.

---

> ### Author Rebuttal · Authors · 2025-03-31
>
> We sincerely thank the reviewer for the time to read our paper. We address your feedback point by point below.
>
> **W1:** Overclaim of lossless.
>
> By "lossless", we primarily emphasize the continuous treatment of visual signals compared to discrete tokens. We will clarify this in the revised version of the paper.
>
> **W2&Q1:** Training efficiency.
>
> Pretraining a unified multimodal model from scratch is highly computationally intensive. For example, Show-o-1.3B-base uses 35M image-text pairs and requires training on 48 A100 GPUs for 500k steps, consuming thousands of GPU hours. In contrast, through adaptation of the pre-trained Chameleon, Orthus-7B-base only requires 72 GPU hours, showcasing its super efficiency.
>
> **W3&Q3:** Quantitative evaluations for interleaved image-text generation.
> Thank you for your suggestion. For quantitative analysis, we use GPT-4V to compare the storybook generation quality of Orthus and MM-Interleaved[1] and report the win rate of Orthus in the table below. Results show that Orthus excels in generating logically coherent interleaved image-text with high relevance and we will add this table in the revised version.
>
> [1] MM-Interleaved: Interleaved Image-Text Generative Modeling via Multi-modal Feature Synchronizer
>
> |   | Image quality & consistency | Text quality & continuity | Text-Image alignment |
> |---- | :-----:| :-----:| :-----:|
> |Win rate | 0.72| 0.51 |0.66|
>
> Here's the prompt we use for evaluation:
> ```
> "Please act as an impartial judge and evaluate the quality of the generation story contents provided by two AI assistants. Your job is to evaluate which assistant's generation is better. Your evaluation should consider image quality and consistency. Avoid any position biases and ensure that the order in which the responses were presented does not influence your decision. After providing your explanation, output your final verdict by strictly following this format: \"[[A]]\" if assistant A is better, \"[[B]]\" if assistant B is better."
> ```
>
> **W4&Q2:** Temperature sensitivity and noise management in the diffusion head.
>
> To analyze the sensitivity of temperature, we ablate different choices and compare their performance in the table below. Results indicate that the model performance is robust to variations in this hyperparameter. We adopt the typical DDPM training SNR schedule[1] in the diffusion head without any tuning. More studies on these will be added to the revision.
> |  | T=0.1 | T=1 | T=10 |
> |---- | :-----:| :-----:| :-----:|
> |POPE | 78.2 |78.7  |78.0|
>
> [1] Denoising Diffusion Probabilistic Models, NIPS 2020.
>
> **S3:** Detailed information on the dataset selection.
>
> The details of training data can be found in line322, line355-358, and line622.
>
> **S4:** Explain for d=30.
>
> This is the model depth of the baseline method SD3.
>
> **Q4:** Robustness in imbalanced or partially missing interleaved inputs.
>
> Orthus demonstrates generalization ability for unseen interleaved inputs. As shown in Figure 3, it successfully completes the task even when provided with unseen interleaved formats during training.
>
> **Q5:** Potential limitations.
>
> Orthus exhibits slightly lower image generation efficiency than Chameleon due to the introduction of the diffusion head. Potential solutions include adopting a faster diffusion sampler or consistency distillation.
>
> **Essential References Not Discussed:** Latest benchmarks including MMIE and OpenING.
>
> Thanks for your suggestion. We primarily focus on fine-tuning Orthus on downstream domain-specific tasks such as image editing and storybook generation to demonstrate its effectiveness in modeling interleaved image-text. We also finetune Orthus on the open-source [WebSight](https://huggingface.co/datasets/HuggingFaceM4/WebSight) dataset to generate webpages (HTML code equipped with images) based on text prompts. Here is an [anonymous link](https://0x0.st/828x.html) showcasing results on this downstream application for your reference. Regarding benchmarks such as MMIE and OpenING, they primarily evaluate interleaved image-text understanding and generation capabilities in more general domains, which are not our main focus. Nevertheless, we evaluate Orthus on MMIE and present the results in the table below.
>
> |  | Orthus | MiniGPT-5 | GILL |
> |---- | :---:| :---:| :---:|
> |MMIE-PBL | 0.568 | 0.551  | 0.576|
>
> Despite demonstrating reasonable performance, Orthus sometimes generates text-only responses or replies with "Sorry, I cannot assist with that." due to the lack of instruction tuning or downstream SFT. We plan to finetune Orthus with more diverse and complex interleaved datasets  (e.g., MMC4) for further evaluation afterward. Moreover, OpenING evaluates interleaved generation methods through pair-wise comparisons. Since other models' results are not yet open-source, evaluation is infeasible now. We will conduct tests once they are released.
>
> Given these improvements and additional experiments to your initial concerns, we hope you would like to raise your score.

---

> > ### Comment · Reviewer_ofPN · 2025-04-03
> >
> > Thank you for the detailed results, which offer valuable insights. Since OpenING also provides subjective scoring tools using GPT-4o, it would be helpful to see some results based on that as well. However, it is completely understandable if time constraints prevent the authors from conducting these additional experiments.

---

> > > ### Author Response · Authors · 2025-04-07
> > >
> > > We sincerely appreciate your acknowledgement of our detailed results and the valuable insights they provide. While our primary focus is on finetuning Orthus for downstream tasks with quantitative analysis to demonstrate its effectiveness in modeling interleaved image-text, we conduct additional experiments on the Interactive Visual Design of OpenING and report subjective scores evaluated by GPT-4o for your reference.
> > >
> > > |  | Orthus | Show-o | NExT-GPT | MiniGPT-5 | GILL | SEED-X |
> > > |---|:--------:|:--------:|:------:|:-------:|:------:|:------:|
> > > |OpenING-IVD $\uparrow$ | 6.3| 5.1 | 5.2 |5.3 |6.2 |8.0|
> > >
> > > We hope these extended evaluations help to address your feedback. If all your concerns have been resolved, we would be grateful if you would consider raising your score. If you have any additional questions or suggestions, we would be happy to have further discussions. Thank you so much!

---

### Official Review · Reviewer_iJo1 · 2025-03-14

**Overall Recommendation:** 3

**Summary:**

This paper introduces an architecture to conduct unified multimodal understanding and generation. Specifically, it introduces a dedicated diffusion head to generate continuous visual token during image generation. By doing so, the proposed approach can enable native image generation with LLM at a low training cost.

## update after rebuttal
Given the clarification and experimental results provided by the authors, which largely resolved my concerns, I would like to keep my score as weak accept.

**Claims And Evidence:**

Yes, the claims are supported by proper evidence.

**Essential References Not Discussed:**

All related references are properly discussed.

**Experimental Designs Or Analyses:**

Yes, I have checked soundness of the experimental designs.

**Methods And Evaluation Criteria:**

Yes, the proposed methods and evaluation criteria make sense for the problem.

**Other Comments Or Suggestions:**

None.

**Other Strengths And Weaknesses:**

Strength:
1) The proposed design achieves impressive results on simultaneous visual understanding and generation.
2) The proposed architecture has the potential to achieve good visual quality in a relatively efficient manner.
3) The proposed approach generalises well to interleaved text-image generation.

Weakness:
1) A numerical comparisons on the inference latencies of the proposed approach and other baseline methods should be provided.
2) With the current architecture with VAE, it might be still challenging to scale to higher resolution for generated images.

**Questions For Authors:**

1. See weakness (1), (2). My concern is still mainly focusing on the efficiency of the proposed approach and the ability to scale to higher resolution.

2. Following fine-tuning for interleaved text-image generation tasks (e.g., image editing, storytelling), is it possible for the model’s visual understanding and text-to-image generation performance to remain at the same level as before fine-tuning?

**Relation To Broader Scientific Literature:**

The proposed architecture could inspire future works on architecture design of unified visual understanding and generation model. Although previous works such as Transfusion achieves relatively good performance, the inference cost is too high. With the design of the dedicated diffusion head, the generation efficiency could be better.

**Theoretical Claims:**

Not applicable.

---

> ### Author Rebuttal · Authors · 2025-03-31
>
> Thank you for the positive feedback and useful suggestions! We are glad you thought our proposed design achieved impressive results. We address your concerns point by point below.
>
> **W1&Q1:** Numerical comparisons on the inference latencies of the proposed approach and other baseline methods.
>
> Thanks for your suggestion. We estimate the practical inference time (sec/image) for generating images at the resolution of 256 with Orthus-7B and Chameleon-7B under the hugging face inference framework. Transfusion-7B employs bidirectional attention and functions as a DiT for image generation. This design prevents the use of kv-cache optimization, resulting in high latency at each diffusion sampling step. However, when the number of sampling steps is significantly lower than the number of generated image tokens, acceleration over AR frameworks could be possible. We will test the exact inference time of Transfusion-7B and provide a detailed comparison once it is open-sourced.
>
> | **Sampler**       |  **Orthus** | **Chameleon** |
> | -------| :---------:| :---------:|
> | DDIM 100 steps | 38.6 | 14.2 |
> | DDIM 50 steps | 26.2 | 14.2 |
> | DPM-Solver++ 10 steps | 16.8 | 14.2|
>
> As shown in the table above, Orthus exhibits slightly higher latency than Chameleon, due to the diffusion head (41M parameters) requiring ~0.001s per forward pass. **However, this latency can be significantly reduced by decreasing the number of forward passes with advanced diffusion samplers or consistency distillation (requiring only 1–4 steps for sampling [1]) and by reducing the per-step computation cost with a more lightweight MLP.** It is also worth noting that the transformer backbone of Orthus and Chameleon shares the same architecture as LLaMA, which is already supported by the vLLM serving framework (achieving up to 1k tokens/s, enabling image generation in less than 1s). This allows for further acceleration of inference speed. We will include these analyses in the revised version and will actively implement these improvements in the future.
>
> [1]Song, Yang, et al. Consistency Models. ICML 2023.
>
> **W2&Q2:** The ability to scale to higher resolution.
>
> Thanks for pointing this out. Currently, Orthus supports image generation up to a resolution of 512, which is already advanced among unified multimodal models (e.g., 256 for Transfusion, 256 and 512 for Show-o, and 384 for Janus).
>
> For even higher resolutions, such as 1024×1024, we have checked that our used VAE is capable of encoding and decoding them. Thus, the main challenge lies in the increased sequence length required for higher-resolution images. To address this, we can interpolate the RoPE positional embeddings of our transformer, following [1], and fine-tune the model to extend its capability for processing and generating longer sequences. We can compress the sequence length in attention computation with KV token compression. Alternatively, we can use another VAE with a higher spatial compression rate, such as a partitioned VAE[2], to reduce sequence length. We will explore this in future work.
>
> [1] Infinity: Scaling Bitwise AutoRegressive Modeling for High-Resolution Image Synthesis. CVPR 2025.
>
> [2] Diffusion-4K: Ultra-High-Resolution Image Synthesis with Latent Diffusion Models. arXiv:2503.18352.
>
> **Q3:** Following fine-tuning for interleaved text-image generation tasks (e.g., image editing, storytelling), is it possible for the model’s visual understanding and text-to-image generation performance to remain at the same level as before fine-tuning?
>
> It is challenging to maintain the same level of performance after downstream SFT due to the common issue of catastrophic forgetting[1,2]. We conduct a toy experiment to mitigate this by incorporating the original instruction-tuning data during storytelling finetuning, yet still observe a performance drop of approximately 10% on the POPE benchmark. This can possibly be alleviated by better data mixture and regularization strategies.
>
> [1] An empirical investigation of catastrophic forgetting in gradient-based neural networks. ICLR 2014.
>
> [2] Overcoming catastrophic forgetting in neural networks. PNAS 2017.
>
> Given these clarifications and improvements to your initial concerns, we hope for your reconsideration in raising your score.

---

> > ### Comment · Reviewer_iJo1 · 2025-04-08
> >
> > Thanks for the clarification! I will keep my score of weak accept.

---

> > > ### Author Response · Authors · 2025-04-09
> > >
> > > Thank you for your valuable feedback！

---

### Official Review · Reviewer_eJsk · 2025-03-16

**Overall Recommendation:** 3

**Summary:**

This paper proposes Orthus, an interleaved image-text generation model with modality-specific heads. Orthus shows that language model heads for discrete tokens and diffusion heads for continuous image generation can work together. By fine-tuning from Chameleon, Orthus obtain good performance on both image understanding and generation.

**Claims And Evidence:**

N/A

**Essential References Not Discussed:**

Missing some important baseline, including VILA-U[1], Janus[2].


[1] VILA-U: a Unified Foundation Model Integrating Visual Understanding and Generation.
[2] Janus: Decoupling visual encoding for unified multimodal understanding and generation.

**Experimental Designs Or Analyses:**

N/A

**Methods And Evaluation Criteria:**

N/A

**Other Comments Or Suggestions:**

1. In the ablation study, are the data all the same for understanding only and generation only tasks compared with unified training?

**Other Strengths And Weaknesses:**

Strength:
1. The idea is easy to follow and understand.
2. The model shows good performance in image editing, generation, and understanding.

Weakness:
1. The idea is relatively not novel enough, just combining diffusion head and lm head, which shows limited insight into unified image understanding and generation.

**Questions For Authors:**

N/A

**Relation To Broader Scientific Literature:**

Show the effectiveness of combining the diffusion head proposed in MAR and token prediction.

**Theoretical Claims:**

N/A

---

> ### Author Rebuttal · Authors · 2025-03-31
>
> Thank you for the positive feedback and useful suggestions! We address your concerns point by point below.
>
> **Weakness:** The idea is relatively not novel enough, just combining diffusion head and lm head, which shows limited insight into unified image understanding and generation.
>
> - We would like to emphasize that **our combination of diffusion head and lm head leads to a unified model for flexible interleaved image-text modeling, mitigating the issues of existing fully AR and AR-diffusion mixed models.** Such abilities are important for downstream tasks like in-context editing and storybook generation, as proven by the recent GPT-4o, but have been inadequately explored in research community.
> - The incorporation of diffusion head enables the processing of continuous visual signals. And, experiment results in Table 2&3 show that lossless continuous visual signal benefits both image understanding and generation. Namely, we yield the insight that the diffusion head can be essential when we need a unified multimodal generation model.
>
> **Missing baselines:** Missing VILA-U and Janus as baselines.
>
> Sorry for the missing comparisons and we include them below:
> - We first clarify that VILA-U and Janus focus solely on image understanding and generation, lacking support for interleaved image-text generation, where Orthus demonstrates strong capabilities. Additionally, Janus decouples visual understanding and generation by using separate encoders, limiting its flexibility in handling interleaved data—where images must be generated and understood simultaneously.
> - We add a comparison with VILA-U and Janus on visual generation and understanding in table below. As shown, Orthus generates images with higher human preference scores at a resolution of 512. For visual understanding, while Orthus still lags, this may be attributed to training solely on the LlaVa-v1.5-mix-665k dataset, whereas the baselines leverage much more high-quality instruction-tuning data [1,2,3]. We plan to incorporate these datasets to further enhance performance.
>
> | **Model**        | **Res.**       | **GenEval $\uparrow$**                | **HPSv2$\uparrow$**           | **POPE$\uparrow$**                   | **MME$\uparrow$**  | **GQA$\uparrow$**  |
> |-------------- |---------------------------|:-------------------------:|:-------------------------:|:--------------------------:|:-------:|:-------:|
> | Orthus   | 512     | 0.58                     | 28.2                     | 79.6                     | 1265.8 | 52.8  |
> | VILA-U | 256    | 0.40                     | 25.3                     | 83.9                     | 1336.2 | 58.3  |
> | Janus | 384                         | 0.61                     | 27.8                     | 87.0                     | 1338.0 | 59.1  |
>
> We will include these comparisons in the revised version. Thank you for your valuable suggestions.
>
> [1] Kvqa: Knowledge-aware visual question answering. AAAI 2019.
>
> [2] Llava-onevision: Easy visual task transfer. TMLR 2025.
>
> [3] Screenqa: Large-scale question-answer pairs over mobile app screenshots. arXiv:2209.08199.
>
> **Question:** In the ablation study, are the data all the same for understanding-only and generation-only tasks compared with unified training?
>
> Yes, for understanding-only and generation-only tasks, we use the same understanding and generation data as in the unified training setting.
>
> Given these reclarifications and improvements in response to your initial concerns, we hope you would like to raise your score.

---

### Decision · Program_Chairs · 2025-05-01

**Decision:**

Accept (poster)

**Comment:**

This paper presents Orthus, an interleaved image-text generation model with modality-specific heads. After rebuttal, it received scores of 3334. All the reviewers are generally happy about the paper, commenting that (1) the idea is easy to follow and understand, (2) the model shows good performance in image editing, generation, and understanding, it also generalizes well to interleaved text-image generation. The newly reported experimental results during rebuttal is also helpful to further support the paper’s contributions.

On the other hand, I also agree that Orthus does not perform that well in multimodal understanding, which may be limited by its relatively weak vision embedding module. Also, the benchmarks reported for multimodal understanding is a bit limited. For example, most of the popular benchmarks for testing text-rich image understanding are not reported, possibly due to performance is not strong on them.

Overall, given the positive feedback from the reviewers and the importance of the topic, the AC agrees that the paper is in a good shape, and would like to recommend acceptance of the paper. The authors are highly encouraged to discuss the limitation of the methods in the revision as well.